# Exposure, hazard, and vulnerability all contribute to *Schistosoma haematobium* re-infection in northern Senegal

Andrea J. Lund[1]*, Susanne H. Sokolow[2,3], Isabel J. Jones[2], Chelsea L. Wood[4], Sofia Ali[5], Andrew Chamberlin[2], Alioune Badara Sy[6], M. Moustapha Sam[6], Nicolas Jouanard[6,7], Anne-Marie Schacht[6,8], Simon Senghor[6], Assane Fall[6], Raphael Ndione[6], Gilles Riveau[6,8], Giulio A. De Leo[2,3], David López-Carr[9]

**1** Emmett Interdisciplinary Program in Environment and Resources, Stanford University, Stanford, California, United States of America, **2** Hopkins Marine Station, Stanford University, Pacific Grove, California, United States of America, **3** Woods Institute for the Environment, Stanford University, Stanford, California, United States of America, **4** School of Aquatic and Fishery Sciences, University of Washington, Seattle, Washington, United States of America, **5** Stanford University, Stanford, California, United States of America, **6** Centre de Recherche Biomédicale–Espoir Pour La Sante, Saint Louis, Sénégal, **7** Station d'Innovation Aquacole, Saint Louis, Sénégal, **8** University of Lille, CNRS, INSERM, CHU Lille, Institut Pasteur de Lille, Center for Infection and Immunity of Lille, Lille, France, **9** Department of Geography, University of California, Santa Barbara, CA, United States of America

* andrea.janelle.lund@gmail.com

**Data Availability Statement:** Individual- and household-level data needed to replicate the analysis cannot be shared publicly because of IRB

## Abstract

### Background

Infectious disease risk is driven by three interrelated components: exposure, hazard, and vulnerability. For schistosomiasis, exposure occurs through contact with water, which is often tied to daily activities. Water contact, however, does not imply risk unless the environmental hazard of snails and parasites is also present in the water. By increasing reliance on hazardous activities and environments, socio-economic vulnerability can hinder reductions in exposure to a hazard. We aimed to quantify the contributions of exposure, hazard, and vulnerability to the presence and intensity of *Schistosoma haematobium* re-infection.

### Methodology/Principal findings

In 13 villages along the Senegal River, we collected parasitological data from 821 school-aged children, survey data from 411 households where those children resided, and ecological data from all 24 village water access sites. We fit mixed-effects logistic and negative binomial regressions with indices of exposure, hazard, and vulnerability as explanatory variables of *Schistosoma haematobium* presence and intensity, respectively, controlling for demographic variables. Using multi-model inference to calculate the relative importance of each component of risk, we found that hazard ($w_i = 0.95$) was the most important component of *S. haematobium* presence, followed by vulnerability ($\sum w_i = 0.91$). Exposure ($\sum w_i = 1.00$) was the most important component of *S. haematobium* intensity, followed by hazard ($\sum w_i = 0.77$). Model averaging quantified associations between each infection outcome and indices of exposure, hazard, and vulnerability, revealing a positive association between hazard and

privacy safeguards, but can be made available by contacting the IRB at the University of California Santa Barbara (hsc@research.ucsb.edu) and Stanford University (irbnonmed@stanford.edu) with a reasonable request. Data aggregated to the village level can be found in the following Dryad repository: https://doi.org/10.5061/dryad. s7h44j15p. Code used in the analysis is freely available at https://github.com/andjanlund/ehv_ schisto.

**Funding:** The study received funding from the National Science Foundation Couple Natural Human Systems program (BCS-1414102 to SHS and DLC) and the Bill and Melinda Gates Foundation (OPP1114050 to SHS and GADL). AJL was supported by a James and Nance Kelso Fellowship through the Stanford Interdisciplinary Graduate Fellowship (SIGF) program at Stanford University. GADL, SHS and AJL were also partially supported by the National Science Foundation (DEB-2011179 to GADL and Belmont CEH/NSF ICER-2024383 to GADL). IJJ was funded by National Science Foundation Graduate Research Fellowship (#1656518). CLW was supported by the Michigan Society of Fellows at the University of Michigan, a Sloan Research Fellowship from the Alfred P. Sloan Foundation, a UW Innovation Award and a grant from the National Science Foundation (OCE-1829509 to CLW). SA was funded by the King Center on Global Development Field Research Assistant Program. The funders had no role in study design, data collection, interpretation or the decision to submit the work for publication.

**Competing interests:** The authors have declared that no competing interests exist.

infection presence (OR = 1.49, 95% CI 1.12, 1.97), and a positive association between exposure and infection intensity (RR 2.59–3.86, depending on the category; all 95% CIs above 1)

## Conclusions/Significance

Our findings underscore the linkages between social (exposure and vulnerability) and environmental (hazard) processes in the acquisition and accumulation of *S. haematobium* infection. This approach highlights the importance of implementing both social and environmental interventions to complement mass drug administration.

## Author summary

While the impacts of natural hazards tend to be described in terms of social determinants such as exposure and vulnerability, the risk for infectious disease is often expressed in terms of environmental determinants without fully considering the socio-ecological processes that put people in contact with infective agents of disease. In the case of schistosomiasis, risk is determined by human interactions with freshwater environments where schistosome parasites circulate between people and aquatic snails. In this study, we quantified the relative contributions of exposure, hazard, and vulnerability to schistosome re-infection among schoolchildren in an endemic region of northern Senegal. We find that hazard and vulnerability influence whether a child becomes infected, while exposure and hazard influence the burden of worms once infection is acquired. Increasing numbers of worms is known to be positively associated with increasing severity of disease. Our findings underscore the importance of evaluating social and environmental determinants of disease simultaneously; omitting measures of exposure, hazard or vulnerability may limit our understanding of risk.

## Introduction

When a natural hazard like a flood or hurricane is imminent, risk managers and first responders must account for the exposure and vulnerability of affected populations, conditions and processes that are dynamic in both space and time [1]. Precious resources will be misplaced if authorities help families whose homes have not flooded, because those resources will have been diverted away from families who have been exposed to the impacts of the hazard. Similarly, vulnerable families with a reduced capacity to anticipate, respond to, or recover from, the impacts of a natural hazard will require more assistance than less vulnerable families [1]. An effective response to natural disasters, as a result, requires evaluation of where a natural hazard overlaps with exposure, and how vulnerability mediates exposure and its impacts in time and space.

In contrast to the risk of natural disasters, the risk of infectious disease is often defined in terms of its environmental determinants [2–4]. Risk of insect-borne diseases like Lyme disease and malaria are often described in terms of 'entomological risk,' such as the density of infected nymphal ticks (for Lyme disease) [5] or the probability of an infective mosquito bite (for malaria) [6]. Similarly, wildlife biodiversity–representing the size of the natural reservoir of potentially zoonotic pathogens–is used as a proxy for spillover risk [7]. While these measures

reflect important components of pathogen transmission, their focus on the source of the disease-causing agent overlooks the socio-ecological processes that connect those infectious agents with susceptible people, thereby facilitating transmission and determining risk. Environmental metrics represent a single component of infectious disease risk: hazard [6, 8]. Meanwhile, exposure and vulnerability reflect the roles of human behavior and social structures in transforming hazard into risk: exposure describes contact between a person and a pathogen (e.g., exposure to a hazard), while vulnerability determines the extent to which people can anticipate, adapt to, and potentially mitigate the impacts of infection [6, 9].

These three components of risk–exposure, hazard, and vulnerability–operate in concert rather than isolation. They comprise the breadth of socio-ecological conditions that put and keep people in contact with infectious agents of disease. Few studies address all three components simultaneously, but those that do show how human interaction with the environment transforms the presence of a hazard into risk for infection and disease. For example, accounting for both entomological risk (i.e., hazard) and exposure explained Lyme disease incidence in New York when entomological risk alone did not [5]. In East Africa, incorporating social vulnerability and entomological risk into a spatially-explicit risk assessment of malaria showed how different interventions could be targeted to different places with different needs [10, 11]. Across disease systems, insights about the relative contributions of exposure, hazard and vulnerability can inform which interventions to employ to mitigate risk and where, when, and how to implement them.

The risk for schistosomiasis–a disease caused by parasitic trematodes from the genus *Schistosoma*–can also be divided into these three distinct components. Human schistosomes complete a two-host transmission cycle between humans and specific species of freshwater snails. People become infected through contact with freshwater where schistosomes' free-living infective larval stage (called cercariae) emerge from infected snails and burrow into human skin. Schistosome hazard is often described as 'malacological risk,' and quantified using ecological indicators of snail populations within a water source [12–14]. Similarly, exposure to schistosomes is often approximated by studying human water contact behavior because the process of cercariae burrowing into skin upon contact cannot be directly observed [15–17]. Even so, exposure to surface water on its own does not imply risk for infection [18, 19], as the hazard of infected snails in a water source, and the circulation of parasites through the snail population is what renders water contact harmful [6]. Lastly, vulnerability determines whether a person can avoid or mitigate the schistosome hazard in their surroundings, either by reducing exposure to or contamination of water bodies [20, 21], or seeking treatment for infection [22]. The relative disadvantage of certain groups may make risk mitigation difficult, obliging people to remain in contact with parasite-laden water even if they are aware of the risk [6].

Approximately 200 million people are infected with schistosomes worldwide, making schistosomiasis second only to malaria in the global burden of parasitic disease [23, 24]. Schistosome infections are treatable with the antiparasitic drug praziquantel. Since the 1980s, mass drug administration (MDA) campaigns employing praziquantel have been the central strategy of schistosomiasis control, displacing the snail control interventions of the preceding decades [25, 26]. These MDA programs have played an important role in reducing morbidity due to schistosomiasis, but because praziquantel does not prevent new infections, the benefits of MDA have not been realized universally across all contexts [27]. The identification of persistent hotspots–or settings whose prevalence and intensity of infection remains high despite repeated rounds of treatment–has underscored the need for interventions that control transmission and complement the morbidity control achieved by MDA [28–30]. Controlling transmission will require both ecological and social points of intervention, whose identification and

implementation can be informed by understanding the relative contributions of hazard, exposure, and vulnerability in the risk for acquiring and accumulating schistosome infections.

In this mixed methods study, we combined parasitological, ecological, and socio-economic data from 13 villages in the lower basin of the Senegal River to better understand the relative contribution of exposure, hazard, and vulnerability to the burden of urogenital schistosomiasis (caused by *Schistosoma haematobium*) in an endemic region of northern Senegal. In this setting, the construction of a dam and irrigation infrastructure has reduced the region's susceptibility to drought and famine, while simultaneously facilitating schistosome transmission. Prior to dam construction, the transmission of *S. haematobium* occurred at low levels on a seasonal basis, but the disease has since become hyperendemic with perennially high prevalence and intensity of infection [31–33]. Dam construction also coincided with the introduction and rapid spread of *S. mansoni*, an intestinal schistosome, which remains co-endemic with *S. haematobium* [34, 35]. Simultaneously, agricultural livelihoods have shifted from traditional flood recession and rainfed cultivation to more intensively irrigated crops [36]. The social and environmental changes in this setting have collectively affected exposure, hazard, and vulnerability to schistosome infection.

Using these data from this setting, we compared the relative contributions of exposure, hazard and vulnerability as explanatory variables of both presence and intensity of *S. haematobium* infection. We then quantified the magnitude and precision of the association between infection outcomes and indices of each component of risk as well as demographic characteristics, whose importance has been demonstrated in this and other settings [37–39]. Using a mix of survey, interview, and ecological data to assess the relative contributions of exposure, hazard, and vulnerability to schistosomiasis risk, we identify the circumstances under which ecological, infrastructural, and behavioral interventions can complement mass drug administration in campaigns to control and eliminate schistosomiasis.

## Methods

### Ethics statement

This study received approval from the National Committee of Ethics for Health Research from the Republic of Senegal (Protocol #SEN14/33) as well as the Institutional Review Boards of the University of California, Santa Barbara (Protocol #19-17-0676) and Stanford University (Protocol #43130). Children were enrolled in the parasitology study after providing assent alongside written informed consent from the child's parent or guardian. All interview and survey respondents provided verbal informed consent.

### Study area and population

This study used cross-sectional, mixed methods data collected in 2017 and 2018 as part of a longitudinal study of schistosome infection in school-aged (5–15 years) children. Primary data collection included parasitological examinations of urine samples screening for *S. haematobium* infections, survey data from the residents and heads of households where those children reside, and interview and ecological data from water access sites in their corresponding villages. Sixteen villages along the lower Senegal River, its tributaries, and the Lac de Guiers in northwest Senegal were selected for participation in the longitudinal study; these villages were representative of the rural, high-transmission sites common in the region (see appendix of [33] for details on village selection; Fig 1). Briefly, the criteria used to select villages included proximity to freshwater, presence of water access sites, presence of a school with sufficient enrollment in target primary school grades, a non-zero prevalence of self-reported infection, and accessibility in the rainy season [33]. Of these 16 study villages, six were located along the

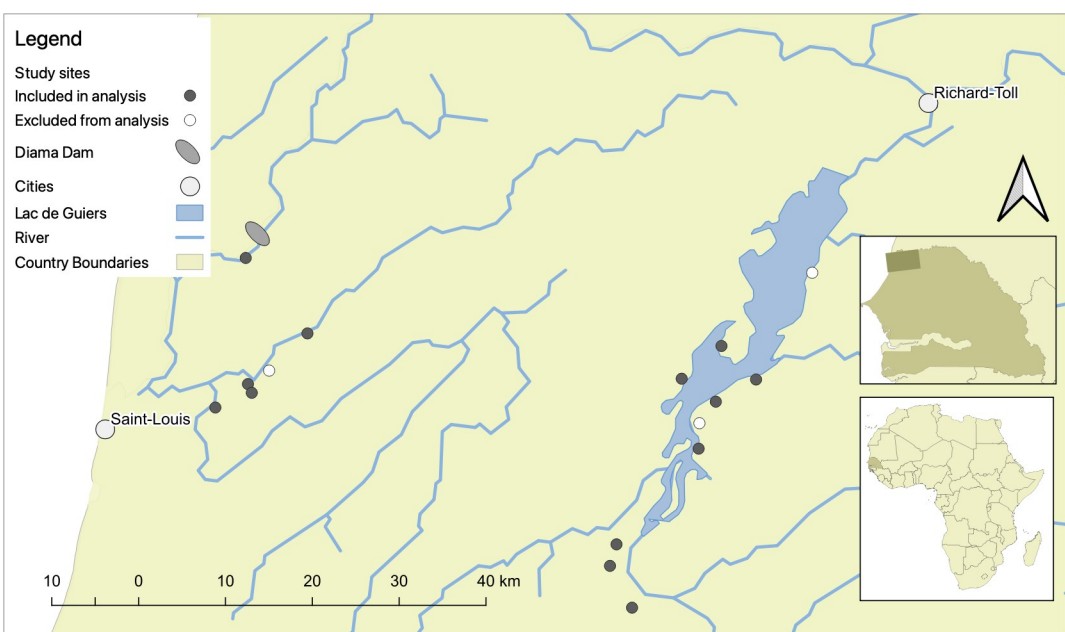

**Fig 1. The lower basin of the Senegal River, where parasitological, socio-economic, and ecological data were collected from 821 school-aged children in 411 households and 24 water access points in 13 villages.** Three villages (shown in white circles) were excluded from this analysis because they were sites of a vegetation removal intervention in 2017 that may have affected hazard indices.

Senegal River and its tributaries while ten were located on the shores of Lac de Guiers. Three of these 16 villages (one on the river and two on the lake) were excluded from this analysis because a vegetation removal intervention took place in these sites in 2017, potentially influencing hazard indices (Fig 1)

School-aged children were recruited from grades 1–3 in each village school, and each student provided urine samples for determining *S. haematobium* infection presence and intensity in January–March 2018. In July 2018, we attempted to reach all the households where students enrolled in the school-based parasitology study lived and administer a questionnaire about household demographics and water contact behavior.

## Data collection

**Parasitology data collection.** Through schools in all 16 villages, a total of 1,480 school-aged children was enrolled at baseline in February–April 2016. Of those, 1,301 remained enrolled during the third year of follow-up in January–March 2018 and successfully produced one urine sample on each of two testing days. After excluding children living in the three villages where the vegetation removal intervention occurred in 2017, parasitological data were available for 821 children in 13 villages (S1 Fig). At baseline and the subsequent two years, urine samples were analyzed by urine filtration for *S. haematobium* infection [40]. Following sample collection, all children were treated at school (with parental consent) with two rounds of praziquantel administration at 40 mg/kg. Because the cross-sectional data used in this study were embedded in a large longitudinal study in the same villages, infection data collected in January–March 2018, which were used as outcome variables in this study, reflect re-infection following treatment over the course of the preceding year [41–43].

**Household survey data collection.** Household-level survey data were collected at the beginning of the rainy season in July 2018, following parasitological data collection for that

year. In all 16 villages, we aimed to reach all households with children enrolled in the school-based parasitology study. Of the 1,301 students enrolled in the parasitology study, we reached the households (n = 524) of 1,216 students (93.5%; S1 Fig). The household survey instrument included six main modules (S1 Table). The modules used in this analysis included: (1) individual-level demographic information, (2) individual-level, two-week recall of contact with surface water for six primary domestic and occupational chores requiring contact with water, (3) average frequency of each water contact activity reported at the household level, (4) socio-economic indicators, such as reported access to improved water and sanitation infrastructure and (5) geographic coordinates of household locations. The collection of water contact data in the survey was designed to elicit information on water contact behavior over the course of a typical week.

**Body surface area interview data collection.** In 15 villages, brief interviews were conducted with village residents, usually at village water access sites in August 2016. In each interview, respondents were asked how much of the body typically comes into contact with water for each of six common water contact activities: (1) washing laundry, (2) washing dishes, (3) collecting water, (4) irrigating fields, (5) washing or watering livestock and (6) fishing from shore [39]. Responses for each activity were registered on a diagram used to measure burn size [44]. Two interviews were conducted in each village–one with an adult man and one with an adult woman–for a total of 30 interviews and 210 activity-specific observations (30 interviews x 7 water contact activities).

**Ecological data collection.** Snail surveys were performed at 32 water access sites in all 16 study villages, the details of which are described elsewhere [33]. Briefly, randomized snail sampling was stratified by three microhabitat types (emergent vegetation, non-emergent vegetation, and open water/mud bottom), such that sampling effort within a site was proportionate to the relative area of each microhabitat. Imagery from Google Earth and an unmanned aerial vehicle (drone) was used to identify site boundaries and designate fifteen random sampling points within each microhabitat. At each point, exhaustive sampling for snails occurred within a quadrat (76.2 cm x 48.26 cm x 48.26 cm; area = 0.3677 square meters). The snails collected from each quadrat were placed into labeled vials and returned to the laboratory for identification and infection screening via shedding and dissection assays. Molecular identification of schistosome cercariae shed from captured snails in the laboratory distinguished human-infecting schistosome species from species that affect non-human animals. Data from the 24 water access sites that were not part of the vegetation removal intervention were included in the analysis.

**Data availability.** A village-level summary of all variables used in this analysis is available online [45].

## Data analysis

**Infection metrics.** Infections with *S. haematobium* were determined to be present when any of the samples collected from an individual student contained at least one *S. haematobium* egg. Infection intensity was quantified by taking the median egg count (per 10 mL urine) across the samples collected from each student at both parasitological sampling visits. Medians were taken because the distributions of infection intensity were strongly right-skewed and included a large number of zeroes, reflecting a negative binomial distribution.

**Exposure indices.** Exposure indices were calculated using data from three sources: (1) individual- and household-level survey data, (2) body surface area interviews and (3) a previous study of water contact behavior in a similar setting [39]. These indices aggregated reported water contact across the six water contact activities described above, including in the body

surface area interview form and the water contact module in the household survey. Whether or not a person performed a given activity in the preceding two weeks was recorded as binary individual-level variables in the household survey. The average frequency of each activity was reported at the household level in the survey, while duration and extent of contact were drawn from literature-based estimates and body surface area interviews, respectively (S2 Table). Exposure indices based on these activity-specific data incorporated varying degrees of complexity, including weekly frequency-only estimates, weekly time exposed and weekly time exposed adjusted for mean body surface area exposure.

Two indices measured frequency of contact with water. One index [*F (raw)*] was a categorical variable derived directly from a single survey question, where each child was asked to recall the number of visits made to a water access point in the preceding seven days. The second [*F (sum)*] was a numeric variable derived from activity-specific measures collected through the survey at the individual and household levels. Individual-level water contact data indicated whether each member of the household had performed each activity in surface water (e.g., lake, river, irrigation canals) in the preceding two weeks. Dichotomous responses were registered for all members of the household. Response categories for frequency of each water contact activity were collected at the household level (e.g., one value per household) and converted to numeric values representing times per week (e.g., a reported frequency of once per day was converted to a numeric value of 7 times per week). The numeric values of household-level activity frequency were multiplied by indicators of individual-level activity to generate estimated weekly contacts by activity for all children in the data set:

$$F(sum) = \sum_{activity}(activity\ performed) * (frequency) \tag{1}$$

Total time exposed per week (*FD*) was calculated by multiplying the weekly frequency of an activity by the estimated duration of that activity, as calculated from direct observations in another study in a similar setting [39]. The product of frequency and duration was then summed across all activities for each child to generate an estimated total time (in minutes) of contact with water over the course of a week.

$$FD = \sum_{activity}(activity\ performed) * (frequency) * (duration) \tag{2}$$

Total exposure time per week was then adjusted by estimated body surface area in contact with water for a given activity (*FDB*). We used data from body surface area interviews (S2 Table) to estimate how much of the body came into contact with water for each activity. For each observation, the percent of body surface area (%BSA) was summed for each body segment reported as exposed for that activity using published values for burn size estimation [44]. Mean %BSA for each activity was calculated across all interview respondents. Activity-specific estimates of body surface area exposed (in square meters) were calculated by multiplying mean %BSA by an estimated total body surface area for children under 14 years of age [46]. Estimated body surface area exposure required for each activity was multiplied by the total time exposed for that activity, giving a BSA-adjusted time per week.

$$FDB = \sum_{activity}(activity\ performed) * (frequency) * (duration) * (body\ surface) \tag{3}$$

**Hazard indices.** Hazard indices were calculated from ecological field data collected at the 24 water access points in the same 13 villages where parasitology and household survey data were collected. A previous study in this setting found that area of suitable snail habitat (e.g., area covered by non-emergent aquatic vegetation) was the best environmental predictor of presence and intensity of *S. haematobium* infection [33]. We developed six hazard indices based on area of snail habitat, which varied by (1) season, (2) spatial relationship and (3) level

of data aggregation. Two hazard variables captured the area of suitable snail habitat (1) for the peak transmission season preceding parasitological data collection for each water access point (July–August 2017; *peak*) and (2) summed across the three ecological data collection periods in the year preceding parasitological data collection (May–June 2017, July–August 2017 and January 2018; *year*). Summing habitat area across the year represents an integrated hazard metric across the three dominant climatic seasons in the region. We primarily used data from the water access site nearest to each household, based on straight-line distance, but two additional hazard variables captured the spatial relationship between households and the nearest water access sites by scaling *peak* and *year* hazard indices by the distance between a child's household and the nearest water access site. Two final hazard variables aggregated *peak* and *year* indices to the village level, where the area of suitable snail habitat during the transmission peak and the entire year were summed across all water points in a village.

**Vulnerability indices.** Vulnerability indices were calculated from the household survey data and focused on access (or lack thereof) to water and sanitation infrastructure at the household level (S1 Table). Two survey questions concerned water sources: (1) what is the principal source of water used by members of the household for drinking? and (2) what is the principal source of water used for doing laundry? Responses to these two questions were used to calculate the extent to which each household depended on surface water for household needs (e.g., surface water is used for neither drinking water nor laundry; surface water is used for either drinking water or laundry; or, surface water is used for both drinking water and laundry). This variable was also dichotomized (e.g., a household used surface water for any activity versus no activities).

Two additional survey questions concerned sanitation infrastructure: (1) what type of toilet is primarily used by members of this household? and (2) is this toilet shared with other households? Responses to these two questions were used to calculate a three-level variable describing sanitation access (e.g., private toilet, shared toilet, no toilet). A dichotomous version of this variable was also calculated (e.g., private toilet versus shared or no toilet). This dichotomous categorization was chosen because few households reported having no toilet.

A final index of vulnerability was an index of socio-economic status (SES) derived from data on durable assets and household conditions reported in the household survey [47, 48]. Principal components analysis was performed using the *prcomp()* function in the *stats* package (version 3.5.1) in R [49]. Loadings for individual variables in the first principal component were used to calculate a numeric socio-economic score for each household. Numeric scores were then divided into quintiles.

**Statistical analysis.** We used mixed-effects logistic and negative binomial regression models to assess exposure, hazard, and vulnerability indices as explanatory variables of *S. haematobium* infection presence (i.e., infected versus uninfected) and intensity (i.e., number of eggs detected per 10mL urine), respectively. In all models, we controlled for key social and demographic covariates: age and sex of the child, village location (i.e., river versus lake) and village population size. We also included random intercepts (households nested within villages) to account for the hierarchy in the data. Mixed effects logistic regression was performed using the *lme4* package (version 1.1–21) [50], while mixed effects negative binomial regression was performed using the *glmmTMB* package (version 1.0.1) [51] in R (version 3.5.1) [49].

We used information-criterion (IC)-based model selection and multi-model inference to determine the relative importance of each component of risk [52]. For each outcome, we first fit a set of exposure-only, hazard-only, and vulnerability-only models. The Akaike Information Criterion (AIC) was used to discern the best-fitting index (or indices) of each component of risk. We retained all variables in exposure-only, hazard-only and vulnerability-only models whose ΔAIC was less than 2 across the models in a given group [52]. This subset of variables

was then used to fit models that included various combinations of the three components of risk: (1) exposure and hazard (E-H), (2) exposure and vulnerability (E-V), (3) hazard and vulnerability (H-V) and (4) all three: exposure, hazard, and vulnerability (E-H-V). Because fewer indices of exposure, hazard and vulnerability were used in these combination models of infection presence (based on AIC for exposure-, hazard-, and vulnerability- only models), we fit a set of 28 candidate models with the presence of *S. haematobium* as the outcome and 47 candidate models with intensity of *S. haematobium* re-infection as the outcome. Regression diagnostics indicated no problem with multicollinearity in models combining indices of exposure, hazard, and vulnerability (S2 Fig).

For both sets of models, we calculated AIC, ΔAIC and Akaike weights ($w_i$) [52]. The sum of Akaike weights across the exposure, hazard and vulnerability components of risk was used to quantitatively assess their relative importance across the full set of candidate models for both presence and intensity outcomes [52]. Additionally, for each index of exposure, hazard and vulnerability, we calculated model-averaged estimates of the magnitude and precision of the association with both infection outcomes using the *AICcmodavg* package in R [53]. Odds ratios (OR) and rate ratios (RR) with 95% confidence intervals (95% CIs) were estimated for presence and intensity outcomes, respectively.

## Results

### Characteristics of the study population

Of the 821 school-aged children with complete information across the parasitological, survey and ecological data sets, the mean age was 9.3 years and approximately half (414/821; 50.4%) were male (Table 1). On average, these children lived within 521 meters of a water access point on the Senegal River, a tributary, or the Lac de Guiers, which contained between zero and 149,775 square meters of suitable snail habitat in the peak transmission season. Approximately half of the households where these children lived relied on surface water collected from water

**Table 1. Demographic characteristics of the overall study population, and stratified by sex.**

| Variable | Overall | Males | Females |
|---|---|---|---|
| Observations [n] | 821 | 414 | 407 |
| Age (years) [mean (SD)] | 9.3 (3.1) | 9.2 (2.2) | 9.3 (3.8) |
| Distance to water point (m) [mean (SD)] | 521.6 (505.3) | 522.2 (530.6) | 521.0 (475.5) |
| Water point habitat area (m²) [mean (range)] | 38,857 (0–149775) | 38,898 (0–100,361) | 38,815 (0–149,775) |
| Surface water dependence [n (%)] | | | |
| Neither drinking nor laundry | 400 (48.7) | 211 (51.0) | 189 (46.4) |
| Either drinking or laundry | 265 (32.3) | 129 (31.2) | 136 (33.4) |
| Both drinking and laundry | 156 (19.0) | 74 (17.9) | 82 (20.1) |
| Sanitation facility [n (%)] | | | |
| Private toilet | 679 (82.7) | 342 (82.6) | 337 (82.8) |
| Shared toilet | 134 (16.3) | 68 (16.4) | 66 (16.2) |
| No toilet | 8 (0.9) | 4 (0.9) | 4 (1.0) |
| Asset-based wealth index [n (%)] | | | |
| 1st quintile (Lowest) | 133 (16.2) | 76 (18.4) | 57 (14.5) |
| 2nd quintile | 116 (14.1) | 57 (13.7) | 59 (14.5) |
| 3rd quintile | 150 (18.3) | 82 (19.8) | 68 (16.7) |
| 4th quintile | 201 (24.5) | 100 (24.2) | 101 (24.8) |
| 5th quintile (highest) | 221 (26.9) | 99 (23.9) | 122 (30.0) |

**Table 2. Descriptive analysis of outcome variables (presence and intensity of *S. haematobium* infection) in the overall study population and stratified by sex.**

| Variable | Overall | Males | Females |
|---|---|---|---|
| Observations [n] | 821 | 414 | 407 |
| Infection presence [infected n (%)] | 548 (66.7) | 302 (72.9) | 246 (60.4) |
| Infection intensity [GM (SD)] | 4.9 (117.4) | 6.4 (66.8) | 3.7 (152.0) |

GM = geometric mean eggs/10mL urine

access points for at least one household need, and most households (82.6%) reported primarily using private sanitation infrastructure (Table 1).

The burden of *S. haematobium* among this study population was high. The overall prevalence was 66.7% and was slightly higher among boys (72.9%) than girls (60.4%) (Table 2). The geometric mean of *S. haematobium* infection intensity was 4.9 eggs per 10 mL urine (SD = 117.4) overall, 6.4 eggs per 10 mL urine (SD = 66.8) for boys and 3.7 eggs per 10 mL urine (SD = 152.0) for girls (Table 2).

When asked about contact with surface water generally, more than one-third (39.1%) of school-aged children reported no visits to village water access sites in the preceding seven days. Approximately one-quarter (26.7%) visited once or twice a week and the remainder (34.3%) reported more frequent visits (Table 3). When prompted to recall activity-specific contact with surface water (e.g., doing laundry or washing livestock), the average sum of total weekly contacts across all activities was much higher (mean = 8.5 times per week, SD = 10.9). The weekly frequency of some of these activities (e.g., bathing and dish washing) exceeded that reported from the single survey question prompting seven-day recall of all visits to the water access point (Table 3).

Bathing was assumed to involve contact with the entire body surface, while shore fishing (47.6%) and water collection (45.3%) involved the next greatest body surface area in contact

**Table 3. Descriptive analysis of exposure indices calculated from individual-level responses to water contact questions in the household survey, overall and stratified by sex.**

| Variable | Overall | Males | Females |
|---|---|---|---|
| Categorical frequency [n (%)] | | | |
| No visits per week | 321 (39.1) | 145 (33.0) | 176 (43.2) |
| 1–2 visits per week | 219 (26.7) | 121 (29.2) | 98 (24.1) |
| 3–6 visits per week | 68 (8.3) | 33 (8.0) | 35 (8.6) |
| 7 visits per week | 173 (21.1) | 95 (22.9) | 78 (19.2) |
| 7+ visits per week | 40 (4.9) | 20 (4.8) | 20 (4.9) |
| Weekly activity-specific frequency [mean (SD)] | | | |
| Bathing | 4.7 (5.2) | 4.6 (5.1) | 4.8 (5.3) |
| Water collection | 1.4 (3.4) | 1.0 (2.9) | 1.8 (3.9) |
| Dishes | 1.2 (3.3) | 0.1 (1.0) | 2.2 (4.3) |
| Laundry | 0.7 (1.9) | 0.1 (0.7) | 1.3 (2.5) |
| Livestock | 0.3 (1.5) | 0.5 (1.9) | 0.1 (0.9) |
| Irrigation | 0.2 (1.2) | 0.2 (1.2) | 0.1 (1.3) |
| Fishing | 0.1 (0.8) | 0.1 (0.8) | 0.03 (0.7) |
| Numeric frequency [mean (SD)] | 8.5 (10.9) | 6.6 (7.5) | 10.4 (13.2) |
| Total weekly time exposed [mean (SD)] | 87.3 (111.4) | 66.0 (74.6) | 109.0 (135.9) |
| BSA-adjusted weekly time exposed [mean (SD)] | 74.5 (83.4) | 65.6 (71.7) | 83.5 (93.0) |

with water (S2 Table). Irrigation (22.5%) was the activity with the least body surface area exposed (S2 Table). Using the reported duration of activity-specific water contact, we estimated that the school-aged children in this setting spend, on average, 87.3 minutes (SD = 111.4) in contact with surface water per week, an estimate that is higher for girls (mean = 109.0 minutes, SD = 135.9) than boys (mean = 66.0 minutes, SD = 74.6). Adjusting weekly total duration of contact by total body surface exposed during a specific activity reduced this overall estimate slightly to 74.5 $m^2$-minutes, with similar variation between boys (mean = 65.6 $m^2$-minutes, SD = 71.7) and girls (mean = 83.5 $m^2$-minutes, SD = 93.0).

While survey data indicate that dependence on surface water is not evenly distributed across the quintiles of the asset-based SES index (Kruskal-Wallis test, $X^2$ = 14.0, df = 4, P-value = 0.007), we find that those who do depend on surface water for at least one household task are relatively evenly distributed across SES quintiles (S3 Table). For example, almost a third of the households (29.8%) that use surface water for either drinking or laundry are in the highest SES quintile. We find a similar pattern for dependence on surface water for both drinking and laundry: almost a quarter of households (21.8%) in the highest SES quintile rely on surface water for both chores, which is identical to the proportion of households in the lowest SES quintile with the same reliance on surface water (S3 Table).

## Best-fitting indices of exposure, hazard, and vulnerability

Four exposure indices, six hazard indices, and five vulnerability indices were considered in a set of mixed-effects logistic regression models of *S. haematobium* infection presence (n = 28) and a set of mixed-effects negative binomial regression models of *S. haematobium* infection intensity (n = 47; Table 4). The full comparison of model fit, including the Akaike weight ($w_i$)

**Table 4. Summary of the indices of exposure, hazard and vulnerability that were compared in a set of 28 mixed effects logistic regression models of *S. haematobium* infection presence and a set of 47 mixed effects negative binomial regression models of *S. haematobium* infection intensity and the number (%) of models in each set in which each variable appeared.**

| | Variable | Description | Models (n (%)) | |
|---|---|---|---|---|
| | | | Presence | Intensity |
| Exposure | F (raw) | Categorical frequency of visits to water access point in previous seven days, derived from single survey question | 4 (14.2) | 20 (43.5) |
| | F (sum) | Numeric frequency of weekly contacts with water, based on individual-level binary indicators for seven common water contact activities and average household-level frequencies, both of which were collected through the household survey | 4 (14.2) | 1 (2.2) |
| | FD | Weekly time exposed (in minutes) derived from activity-specific survey data, as in *F (sum)*, combined with literature-derived data on duration of exposure [39] | 4 (14.2) | 1 (2.2) |
| | FDB | Weekly time exposed adjusted for mean body surface area exposed for each of seven common water contact activities, data estimated from body surface area interviews | 4 (14.2) | 1 (2.2) |
| Hazard | areaPeak | Area of snail habitat at the water access site nearest to household during peak season | 1 (3.6) | 8 (17.4) |
| | areaYear | Area of snail habitat at the water access site nearest to household summed over year | 1 (3.6) | 8 (17.4) |
| | areaPeak_d | Area of snail habitat at the water access site nearest to household during the peak season, adjusted by the distance between the site and the household | 1 (3.6) | 8 (17.4) |
| | areaYear_d | Area of snail habitat at the water access site nearest to child's household summed over year, adjusted by the distance between the site and the household | 10 (35.7) | 8 (17.4) |
| | areaPeakV | Area of snail habitat summed across all water points in a village during the peak season | 1 (3.6) | 1 (2.2) |
| | areaYearV | Area of snail habitat summed across all water points in a village and across year | 1 (3.6) | 1 (2.2) |
| Vulnerability | surface | Dependence on surface water for household needs derived from survey data: (0) neither laundry or drinking, (1) either laundry or drinking or (2) both laundry and drinking | 10 (35.7) | 1 (2.2) |
| | surfaceYN | Dichotomous version of surface water dependence for any versus no activities | 1 (3.6) | 10 (21.7) |
| | sanitation | Primary sanitation infrastructure used by members of a household: (0) none, (1) shared toilet, (2) private toilet | 1 (3.6) | 10 (21.7) |
| | privateSan | Dichotomous version of *sanitation*, where members of a household use either a private toilet or shared/no toilet | 1 (14.2) | 10 (21.7) |
| | assetIndex | Quintiles of an asset-based SES index created using principal components analysis | 1 (3.6) | 1 (2.2) |

for each individual model, is summarized in S4 Table for presence models and in S5 Table for intensity models. Data in Table 4 summarize each variable used to approximate exposure, hazard, or vulnerability as well as the number of models that variable appeared in for each outcome. For infection presence, all four exposure-only models were within a ΔAIC of 2. For infection intensity, the categorical frequency of water point visits (*F (raw)*) outperformed all three numeric indices of exposure (numeric frequency [*F (sum)*], total time exposed [*FD*] and BSA-adjusted time exposed [*FDB*]). The exposure-only model of infection intensity containing the *F (raw)* index had the lowest AIC of the four exposure-only models, with the ΔAIC for the next best-fitting model of infection intensity far exceeding the threshold of 2 (ΔAIC = 16.52; S5 Table). As a result, we retained all four indices of exposure for models of infection presence, each of which appeared in a total of four of the 28 (14.2%) presence models. In contrast, the categorical frequency of water access site visits was the only index of exposure retained in subsequent models of infection intensity, appearing in a total of 20 of 47 (43.5%) models in that set (Table 4).

A single index of hazard–the area of snail habitat at the water access site nearest to a child's household summed over the three climatic seasons of ecological data collection, adjusted by the distance between the site and the household (*areaYear_d*)–had a ΔAIC ≤ 2 compared to other hazard-only models of *S. haematobium* presence (S4 Table). As a result, *areaYear_d* was the only index of hazard to appear in more than one model of *S. haematobium* presence (Table 4). In contrast, five indices of hazard had a ΔAIC ≤ 2 compared to other hazard-only models of *S. haematobium* intensity (S5 Table). These variables included the area of snail habitat at the water access site nearest to a child's household during the peak transmission season (*areaPeak*) as well as area summed over the entire year (*areaYear*). The distance-adjusted analogs of these variables (*areaPeak_d* and *areaYear_d*) were also retained as was the area of snail habitat summed across all water points in a village and across the full year (*areaYearV*). These five indices of hazard were retained in subsequent models that combined hazard with other components of risk, such that all five appear in multiple models of *S. haematobium* intensity (Table 4).

One index of vulnerability–categorical dependence on surface water (*surface*)–best fit for the *S. haematobium* presence data (e.g., ΔAIC ≤ 2 among vulnerability-only models; S4 Table). This variable was retained for subsequent models that combined components of risk and appeared in ten (35.7%) of the 28 presence models (Table 4). In contrast, binary dependence on surface water (*surfaceYN*) as well as categorical and binary variables for sanitation infrastructure (*sanitation* and *privateSan*, respectively) all had ΔAIC ≤ 2 among vulnerability-only models of *S.haematobium* intensity (S4 Table). Each appeared in ten (21.7%) of the 47 intensity models (Table 4).

### Relative importance of exposure, hazard, and vulnerability

Across these sets of candidate models, no single model emerged as the best-fitting model of presence or intensity of *S. haematobium* re-infection (S4 and S5 Tables). The largest $w_i$ for mixed effects logistic regression models of *S. haematobium* presence was 0.28. A total of five models were classified as having substantial support (ΔAIC < 2) [52]. Four of these five models included all three components of risk while the fifth (and best-fitting model) contained indices of hazard and vulnerability (Table 5)

The largest $w_i$ for mixed-effects negative binomial regression models of *S. haematobium* intensity was 0.08, with 18 models classified as having substantial support [52]. Half (9 /18) of the intensity models with substantial support included indices of all three components of risk. An index of exposure was included in all the best models of re-infection intensity, including

**Table 5. Inclusion of indices of exposure (E), hazard (H) and vulnerability (V) in models of *S. haematobium* presence and intensity models with substantial support (ΔAIC <2).**

| No. | E | H | V | AIC | ΔAIC | $w_i$ | $\Sigma\ w_i$ |
|---|---|---|---|---|---|---|---|
| *Presence models with substantial support (from S4 Table)* | | | | | | | |
| 24 | | X | X | 837.37 | 0 | 0.28 | 0.28 |
| 26 | X | X | X | 838.32 | 0.96 | 0.18 | 0.46 |
| 27 | X | X | X | 838.57 | 1.21 | 0.16 | 0.62 |
| 28 | X | X | X | 838.89 | 1.52 | 0.13 | 0.75 |
| 25 | X | X | X | 839.21 | 1.84 | 0.11 | 0.86 |
| *Intensity models with substantial support (from S5 Table)* | | | | | | | |
| 1 | X | | | 5494.29 | 0 | 0.08 | 0.08 |
| 16 | X | X | | 5494.41 | 0.12 | 0.08 | 0.15 |
| 19 | X | X | | 5494.46 | 0.17 | 0.08 | 0.22 |
| 42 | X | X | X | 5494.46 | 0.17 | 0.07 | 0.30 |
| 22 | X | | X | 5494.49 | 0.20 | 0.07 | 0.37 |
| 18 | X | X | | 5494.86 | 0.57 | 0.06 | 0.42 |
| 43 | X | X | X | 5494.86 | 0.57 | 0.06 | 0.48 |
| 40 | X | X | X | 5494.95 | 0.66 | 0.06 | 0.54 |
| 36 | X | X | X | 5494.98 | 0.68 | 0.06 | 0.59 |
| 21 | X | | X | 5495.21 | 0.92 | 0.05 | 0.64 |
| 17 | X | X | | 5495.30 | 1.01 | 0.05 | 0.69 |
| 20 | X | X | | 5495.50 | 1.21 | 0.04 | 0.73 |
| 41 | X | X | X | 5495.70 | 1.41 | 0.04 | 0.77 |
| 38 | X | X | X | 5495.81 | 1.52 | 0.04 | 0.81 |
| 37 | X | X | X | 5495.95 | 1.66 | 0.03 | 0.84 |
| 39 | X | X | X | 5496.16 | 1.87 | 0.03 | 0.87 |
| 46 | X | X | X | 5496.19 | 1.90 | 0.03 | 0.90 |
| 23 | X | | X | 5496.23 | 1.94 | 0.03 | 0.93 |

X indicates whether a variable from a given component of risk was included in the model; AIC = Akaike information criterion; $w_i$ = Akaike weight; $\Sigma\ w_i$ = sum of Akaike weights across models in a set

an exposure-only model which had the lowest AIC of all models of re-infection intensity (Table 5).

We used the sum of Akaike weights (Σ$w_i$) across each set of candidate models for each outcome to quantify the relative importance of each component of risk (Table 6). We found that hazard was the most important component of risk for re-infection presence in school-aged children (Σ$w_{i\ =}$ 0.95). The relative importance of hazard for *S. haematobium* presence was followed by vulnerability (Σ$w_i$ = 0.91) and then exposure (Σ$w_i$ = 0.66; Table 6). The relative importance of each component of risk differed for models of *S. haematobium* re-infection intensity, where exposure (measured as categorical frequency of contact with surface water) was the most important metric (Σ$w_i$ = 1.00), followed by hazard (Σ$w_i$ = 0.77) and then vulnerability (Σ$w_i$ = 0.63).

**Table 6. Sum of Akaike weights (Σ$w_i$) as a measure of relative importance of the three components of risk for *S. haematobium* infection presence and intensity.**

| | Σ$w_i$ | | |
|---|---|---|---|
| | **Exposure** | **Hazard** | **Vulnerability** |
| Infection presence | 0.66 | 0.95 | 0.91 |
| Infection intensity | 1.00 | 0.77 | 0.63 |

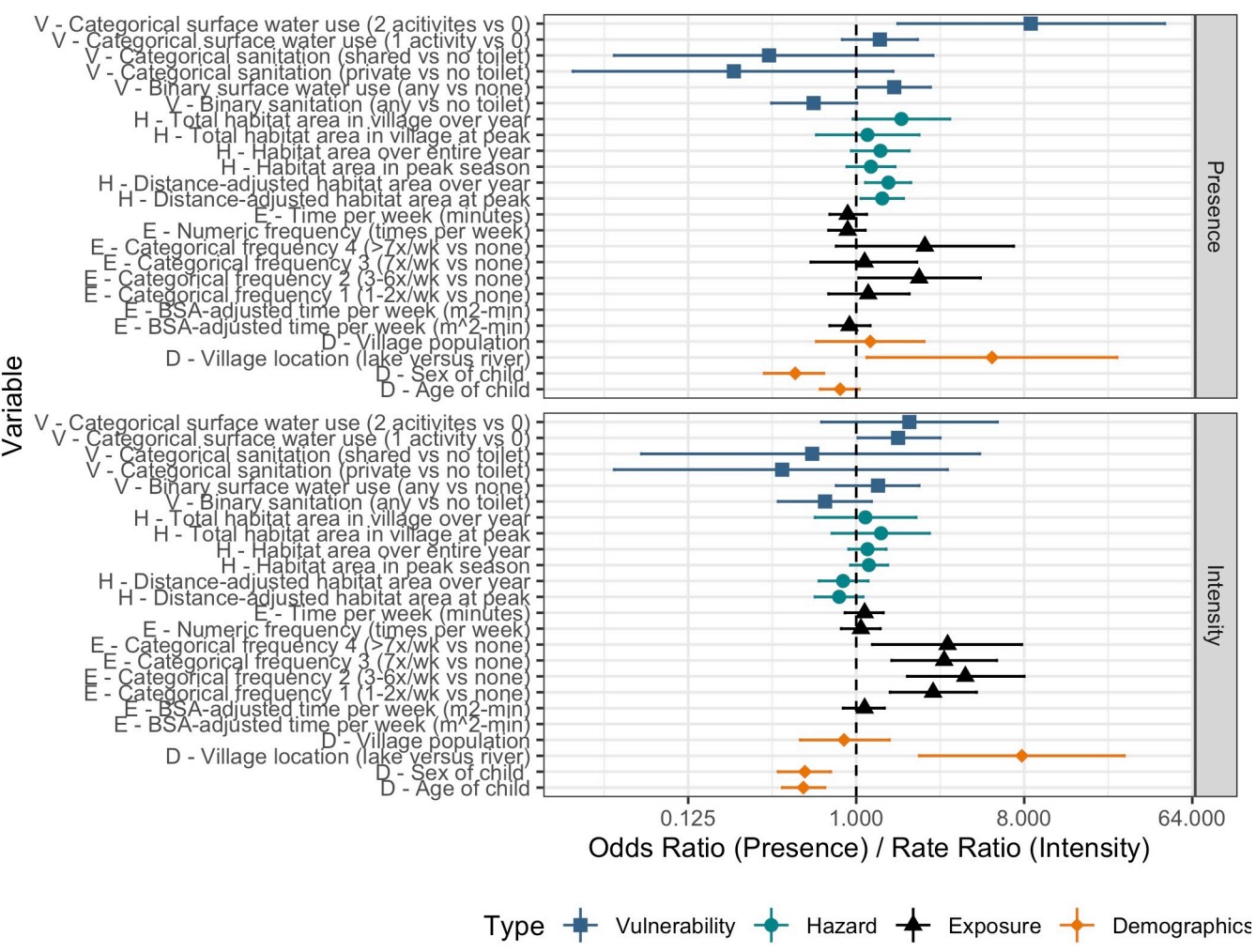

**Fig 2. Model-averaged point estimates and 95% confidence intervals for association between infection outcomes and each component of risk.** Top: Associations between *S. haematobium* infection presence and components of risk are estimated by logistic regression and measured with odds ratios. Bottom: Association between *S. haematobium* infection intensity and components of risk are estimated by negative binomial regression and measured with rate ratios. In both panels, indices of exposure (E) are represented by black triangles, indices of hazard (H) by turquoise circles, indices of vulnerability (V) by blue squares and demographic control variables (D) by orange diamonds.

## Model-averaged estimates

We used model averaging to calculate the magnitude and precision of the association between both infection outcomes and all indices of exposure, hazard and vulnerability that appeared in more than one model across the full set of candidate models and the demographic control variables used in each model (Fig 2). For the infection presence outcome, we found that the odds of infection increased with distance-adjusted area of snail habitat at the nearest water point to a child's household summed over the year preceding infection data collection. The association corresponds with a 45% increase in the odds of infection with each square meter of snail habitat over the course of an entire year, with a confidence interval contained entirely above the null value of 1 (*Distance-adjusted habitat area over year*; OR = 1.49, 95% CI 1.12, 1.97).

Associations between infection presence and indices of vulnerability were less precise, with many confidence intervals including the null value of 1 (Fig 2). When comparing children whose households relied on surface water for multiple household tasks to those who relied on

surface water for no household tasks, we found an almost nine-fold increase in the odds of infection with a confidence interval entirely above the null value of 1 (*Categorical surface water use* [2 activities vs 0]; OR = 8.67, 95% CI 1.67, 45.60). A similar association comparing households' use of surface water for a single activity was also positive, but its confidence interval crossed the null (*Categorical surface water use* [1 activity vs 0]; OR = 1.34, 95% CI 0.84, 2.14).

All levels of the categorical exposure index suggest a positive association with infection presence (categorical frequency [*F (raw)*]), but only one level (*Categorical frequency* [3–6 times/week vs none]) had a confidence interval that did not include the null value of 1 (OR = 2.18, 95% CI 1.03, 4.66). Remaining levels of categorical exposure (1–2 times/week vs none, 7 times/week vs none and >7 times/week versus none) cannot be considered statistically different from the null (Fig 2). All three numeric indices of exposure had point estimates slightly below 1 (e.g., 0.9) with confidence intervals that included in the null value (Fig 2).

Two of the four demographic control variables had significant associations with infection presence (Fig 2). Children living in lake villages were five times more likely to be infected compared to children living in river villages (*Village location*; OR = 5.37, 95% CI 1.14, 25.28). The odds of infection were 54% lower among girls compared to boys (*Child sex*; OR = 0.46, 95% CI 0.33, 0.63) and younger children were less likely to be infected than older children (*Child age*; OR = 0.78, 95% CI 0.63, 0.96). Village population size was not associated with infection presence (*Village population*; OR = 1.16, 95% CI 0.64, 2.10).

For the infection intensity outcome, we found that the median number of eggs detected in urine samples increased for all levels of exposure (*Categorical frequency* [*F (raw)*]; Fig 2). These point estimates ranged from a 2.59- to 3.86-fold increase in egg burden among children reporting any exposure compared to those reporting no exposure. Confidence intervals for all estimates were entirely above the null value of 1 (95% CIs: [*1-2x/week vs none*] 1.52, 4.44, [*3-6x/week vs none*] 1.88, 8.00, [*7x/week vs none*] 1.55, 5.70, [>*7x/week vs none* 1.22, 7.77; Fig 2). The directionality of the associations between infection intensity and indices of vulnerability are like those for infection presence (e.g., above 1 for surface water use, below 1 for sanitation infrastructure), but with point estimates closer to the null and wider confidence intervals that include the null value of 1 (Fig 2).

The measures of association between infection intensity and hazard are mixed: distance-scaled and village-level indices have point estimates less than 1 while raw values of habitat area (i.e., not scaled by distance) for the nearest water point to a child's home are greater than 1. The confidence intervals for all associations between *S. haematobium* infection intensity and hazard indices include the null value of 1. Associations between infection intensity and demographic control variables were similar to the associations described for these variables with presence outcomes (Fig 2).

## Discussion

We found that components of risk describing exposure, hazard, and vulnerability all had some importance in explaining the occurrence of *S. haematobium* in school-aged children in this endemic setting in the lower Senegal River Basin. Based on the sums of Akaike weights calculated across both sets of models, however, hazard and vulnerability were the most important components of *S. haematobium* infection presence, while exposure and hazard were the most important components of *S. haematobium* infection intensity. These differences highlight the distinct factors that influence whether someone becomes infected versus whether they accumulate a large worm burden, with implications for the design of schistosomiasis control and elimination programs and guidelines.

The importance of hazard and vulnerability to the infection presence outcome reflects the conditions that influence whether a person acquires an infection or not. Any exposure may lead to infection, making additional exposure irrelevant to this binary outcome once a person becomes infected. But hazard in the environment, or the number of infected snails emitting cercariae from a certain amount of available habitat, is more likely to increase the risk of infection so long as some exposure is happening. Vulnerability through the household activities that involve regular water contact, like dependence on surface water, for example, ensures some baseline level of exposure.

Similarly, the importance of exposure and hazard to infection intensity reflects the conditions that influence the accumulation of worms. The intensity of infection is theoretically proportional to exposure, with each new adult worm the result of a separate infection event [54]. While this has not always borne out empirically [18, 39], our data agree with the theoretical expectation that more exposure leads to higher intensity infection. Such accumulation of worms is important because it can lead to severe morbidity. A study in Uganda found the prevalence of liver fibrosis to be five times greater in people who had lived in the *S. mansoni*-endemic area since birth compared to those with just a 10-year residency, leading the authors to conclude that the duration of exposure over time is an important risk factor for developing the severe morbidity resulting from prolonged high-intensity infections [55]. Another study in China found that water contact itself was not associated with infection intensity but, similar to our findings, accounting for both water contact and a measure of cercarial risk did [19].

While the recognition that schistosomiasis risk is determined by the convergence of social (exposure and vulnerability) and environmental factors (hazard) is not new [56–59], few studies comprehensively address all three components of risk for schistosomiasis specifically or for infectious diseases generally. Studies employing mathematical models have jointly addressed social and environmental determinants of schistosomiasis [60–62], while empirical studies often focus on one [12, 13, 63] or the other [16, 64, 65]. Our results allow us to think more concretely about how to choose interventions and thereby target schistosomiasis control. Reducing exposure through behavior change interventions that minimize contact with water is likely to contribute more to morbidity control (e.g., reductions in infection intensity) than transmission control (e.g., reductions in infection presence). Reducing vulnerability through the provision and use of piped water infrastructure will give people an alternative to surface water, allowing them to avoid risky exposures and thus, lower infection intensity. Reducing hazard in the environment through snail control interventions will be essential to reducing transmission and the likelihood of acquiring and accumulating infections from the environment. Understanding the interplay between different components of risk in a specific setting can facilitate the design of effective and efficient context-specific control strategies.

In this study, we used primary data sets collected by three distinct teams under the same project in the same study villages: (1) collection, processing and diagnosis of *S. haematobium* infection from urine samples collected from school-aged children on a yearly basis with associated treatment with praziquantel, (2) exhaustive sampling of snail populations and their habitats in the water access sites in all villages, and (3) socio-economic surveys in all the households where school-aged children in the parasitology study lived. All three data streams, combined with data on the spatial relationship between water access sites and surveyed households, were important for explaining patterns of infection in children. Our results underscore the importance of interdisciplinary and mixed methods research when addressing complex socio-environmental problems like the transmission of schistosomiasis and other infectious diseases.

## Measures of exposure, hazard, and vulnerability

We learned how different measures of exposure, hazard and vulnerability fit the infection data. The simplest exposure index, which used seven-day recall of visits to a surface freshwater access point, fit infection intensity data better than did more complex exposure metrics that included information on the reported frequency, duration and extent of contact experienced for specific activities. There are several potential explanations for this finding. This simple index was derived from a single survey question that was posed specifically for the children enrolled in the parasitology study. In contrast, the activity-specific estimates were derived from a survey module that prompted individual-level information from all members of that household. Because of the volume of information that respondents were asked to provide (e.g., whether a person performed each of six water contact activities for all members of the household), the accuracy and precision of the activity-specific responses may have suffered.

Other studies have found weak or no association with exposure measures that account for multiple aspects of exposure. One study in a similar setting in northern Senegal found all of their exposure indices to be correlated with each other but not with the presence or intensity of *S. mansoni* infection [18]. In the study from which we drew data on activity-specific duration of water contact, a relationship between water contact and intensity of *S. mansoni* infection was only detected after accounting for body surface area and the time of day of exposure [39]. In a study of *S. haematobium* infection in Zanzibar, a set of exposure indices that incorporated varying combinations of frequency, duration and body surface area were found to be positively associated with infection, but the strength of the association decreased with the complexity of the exposure index used [66]. Moreover, the distinct ecologies of schistosome species and their intermediate snail hosts and the methods used to measure water contact may confound the true relationship between water contact and infection.

Similarly, malacological hazard data often have weak or no association with infection. In one study in China, a single infected snail was found among 7,000 snails collected in a setting where prevalence of *S. japonicum* infection in people exceeded 25% [67]. Prevalence of human infection of 30–40% has been observed in lake regions of East Africa where snails have been found to occur in very low numbers [68, 69]. An ecological study of snail ecology in Senegal– the one from which ecological data in this study is drawn–found the area of snail habitat to be a superior predictor of human infection compared to seemingly more proximate predictors including density of infected snails [33]. The current study builds on these findings by matching individual-level infection to the water access site nearest to the household (in addition to village-level metrics used in the previous study) and accounting for the spatial relationship between the household and the water access site.

Given the shortcomings of previous studies on water contact and snail ecology, we found that accounting for the spatial relationship between people and the environmental hazard improved the fit of *S. haematobium* presence models. Hazard indices scaled by the distance to the nearest water point fitted the infection presence data better than the other hazard indices, though this was not the case for infection intensity data. Previous studies have found the proximity to water access points to be an important explanatory variable of schistosome infection in endemic settings [66, 70–73]. Another found that water contact was only related to re-infection after accounting for the spatial distance between people and the hazard in their environment [19]. Such proximity may be an important aspect of vulnerability that makes it difficult to avoid exposure to hazardous water.

Similarly, hazard indices that dealt with the site nearest to the child's household fit the infection presence data better than village-level aggregations of snail habitat. This finding contradicts studies that found schistosomiasis to be associated with human behavior and snail

ecology primarily at the community level [74–76]. However, because water contact activity for most people tends to occur at just one or two sites [75, 76] and since distance to water is commonly associated with infection [71, 72], it is possible that the distance-adjusted hazard indices capture the hazard where exposure is most likely to occur for people living in a given household.

## Measures of association between components of risk and infection

We found that no single model best approximated either the presence or intensity of *S. haematobium* infection. Rather, a set of models comprised the evidence for our conclusions. Without a single best-approximating model, we used model averaging to generate point estimates and confidence intervals of the association between our two infection outcomes and all explanatory variables appearing in both model sets.

Associations between demographic control variables and infection were consistent across the two outcomes. These variables–particularly child sex and village location–had the strongest relationship with infection of all variables included in all models. This finding speaks to the importance of demographics in the dynamics of *S. haematobium* infection [77, 78] and the need for analyses that examine heterogeneity across sex and geographic strata [76, 79].

We found that exposure was associated with an increase in both infection presence and intensity, although the confidence interval for the association with infection presence included the null value. Consistent with our relative importance findings, these estimates support the notion that the amount of exposure influences the accumulation of worm burden more than the acquisition of infection. The substantial support we found for vulnerability as an explanatory variable of infection presence may approximate the presence (rather than amount) of exposure. Given the lifespan of adult schistosomes in the human host (3–10 years) [77, 80, 81], a single infection resulting from a single exposure may be present in the body for years until a person is treated.

Even though model-averaging found indices of vulnerability to have wide confidence intervals, there was suggestive evidence of a dose-response between the odds of infection and the number of activities for which a household uses surface water. Children living in households that relied on surface water for multiple activities were more likely to be infected than those whose households relied on surface water for a single activity. While this may be the result of a lack of alternative sources of water (such as wells or piped water), it is also important to note that water contact activities often have social value [82], making the use of surface water for social activities like laundry a matter of preference. It is also possible that available alternative sources of water may not be suitable for particular activities, making surface water preferable in some cases [65, 83].

The asset-based SES index was the poorest index of vulnerability (ΔAIC = 9.48 and 6.40 among vulnerability-only presence and intensity models, respectively), outperformed by indices that represented access to, and use of, water and sanitation infrastructure. It is possible that SES is a relatively poor predictor of infection because there is not much variability in ownership of durable assets across our study population. Similarly, while the household dependence on surface water is not evenly distributed across SES quintiles, we found similar proportions of households in the highest and lowest SES quintiles that use surface water for at least some household needs. It is easy to assume that dependence on surface water will be highly correlated with SES, but that may not be the case when decisions to continue using surface water are made for other reasons. Similarly, because our study focuses on children who go to school and play with each other, the influence of SES on infection outcomes may not be as

pronounced as they would be among adults, for whom occupation and household factors are stronger determinants of exposure. Consequently, SES, as we measured it, may not be a strong determinant of water contact or *S. haematobium* infection.

Finally, our measures of hazard were improved by accounting for snail habitat in the water access site nearest to the household, and in the case of models of *S. haematobium* presence, the best metric accounted for distance between households and the nearest water point. These methodological insights highlight the important role of spatial relationships in schistosomiasis risk and the value of incorporating spatial data and analysis into future research.

## Limitations

The limitations of this study are grounded in the challenge of measuring behavioral and eco-logical phenomena that are dynamic in both space and time. In using multiple methods to measure water contact behavior, specifically, we struck a balance between collecting complete information across the many facets of water contact behavior and collecting data from enough participants to fit statistical models. To do this, we supplemented individual-level information on regular water contact activities for all participants with household-level survey data on frequency of contact, literature-based estimates on duration of contact and interview data on body surface area exposure.

While incorporating frequency, duration and body surface area into exposure indices was meant to triangulate around the different dimensions of water contact, we note that the complex metrics were often outperformed by a simple metric of exposure derived from a single survey question. The poor fit of the complex exposure indices may reflect our use of aggregated data on frequency, duration and extent of exposure to the household and village levels, as they may obscure important individual-level heterogeneities [54, 84]. Ultimately, logistics limited our ability to capture water contact behavior comprehensively and precisely for hundreds of children across more than a dozen villages, each with its own complex landscape of water contact. Because we did not record individual episodes of water contact for each participant, we were also not able to determine the time of day when water contact occurred, a key criterion for exposure based on the chronobiology of cercarial release from snails [85, 86].

The issues implicit in the choice of methods for measuring water contact (e.g., recall and/or social desirability bias for questionnaire-based studies and failure to capture contact at other sites for studies based on direct observations at a single site) may be resolved in future studies by combining questionnaire-based or activity diary methods with the use of wearable GPS data loggers to collect more accurate and precise data on the spatial and temporal dynamics of water contact behavior. Such methods have been used in China [87], Uganda [82] and Camer-oon [88] and have been found to be widely acceptable [89].

Finally, this study examines the convergence of the processes of exposure, hazard, and vul-nerability for just one of the two schistosome species that circulate and cause disease in this set-ting. *S. mansoni* is co-endemic, circulating within different species of snail intermediate hosts with distinct habitat associations. The human behaviors associated with contamination of the environment also differ for *S. mansoni* compared to *S. haematobium*. For these reasons, the particularities of *S. mansoni* transmission may not be governed by the same relationships between exposure, hazard, and vulnerability.

## Conclusion

In summary, we find that hazard, as measured by the area of snail habitat in the water access point nearest a household, and vulnerability, in terms of surface water use and access to sanitation at the household level, contribute most to the acquisition of *S. haematobium*

infections. In contrast, exposure, approximated by a self-reported categorical frequency of contact with surface water per week, and hazard contribute most to the accumulation of *S. haematobium* parasites in school-age children, and thus to infection intensity. Together, our findings underscore the importance of all three components of risk, which act together across time and space to facilitate the acquisition and accumulation of *S. haematobium* infections in an endemic setting with high incidence of post-treatment re-infection. They highlight how interventions to complement MDA may be strategically deployed to reduce the risk of post-treatment re-infection.

While our data are specific to the context in which we collected them, the insights about the interaction among different components of risk and their impact on different metrics of infection may apply in other endemic settings and generate new knowledge about the processes that lead to the occurrence of persistent hotspots. Such an approach is critical to both controlling the severe morbidity of schistosomiasis, reducing the transmission of parasites in the environment and achieving elimination in endemic regions. Moreover, building a more holistic understanding of infectious disease risk will improve our ability to intervene at the point where a source of infection and a susceptible person meet.

## Supporting information

**S1 Checklist. STROBE checklist for cross-sectional studies.**
(DOC)

**S1 Fig. Data flow diagram.** Sample sizes resulting from the merging and cleaning of human data from parasitological and household surveys and removing data from manipulated sites.
(TIF)

**S2 Fig. Correlation plot.** Correlations between all variables used in all models.
(TIFF)

**S1 Table. Household survey items.** Household survey items used in in this analysis, including the module to which an item belonged, the level at which it was measured as well as the wording of and response categories provided for each question.
(PDF)

**S2 Table. Duration and extent of water contact.** Activity-specific values used to estimate duration and extent of water contact. Duration estimates were drawn from published research that took place in a similar setting and extent was estimated from body surface area interviews.
(PDF)

**S3 Table. Household surface water use by socio-economic status.** Frequencies (n (row %)) of household surface water use by quintiles of asset-based index of socio-economic status (SES).
(PDF)

**S4 Table. Presence models.** Full set of candidate models used for multi-model inference of the relative contributions of exposure, hazard, and vulnerability to *S. haematobium* infection presence.
(XLSX)

**S5 Table. Intensity models.** Full set of candidate models used for multi-model inference of the relative contributions of exposure, hazard, and vulnerability to *S. haematobium* infection intensity.
(XLSX)

## Acknowledgments

The authors thank Dr. Seynabou Ndiaye and Dr. Tidiane Ly for their work caring for the health of the children enrolled in the parasitology study. The authors also thank the staff of CRB-EPLS who played critical roles in collecting and processing the parasitology data, the ecological data and the household survey data as well as providing logistical and administrative support of the field work. The authors also thank Ashley Hazel for her helpful comments on an early draft of this manuscript.

## Author Contributions

**Conceptualization:** Andrea J. Lund, Susanne H. Sokolow, Giulio A. De Leo, David López-Carr.

**Data curation:** Andrea J. Lund, Isabel J. Jones, Chelsea L. Wood, Sofia Ali, Andrew Chamberlin, Alioune Badara Sy, M. Moustapha Sam, Nicolas Jouanard, Anne-Marie Schacht, Simon Senghor, Assane Fall, Raphael Ndione.

**Formal analysis:** Andrea J. Lund.

**Funding acquisition:** Susanne H. Sokolow, Chelsea L. Wood, Gilles Riveau, Giulio A. De Leo, David López-Carr.

**Investigation:** Andrea J. Lund, Isabel J. Jones, Chelsea L. Wood, Sofia Ali, Andrew Chamberlin, Alioune Badara Sy, M. Moustapha Sam, Assane Fall, Raphael Ndione.

**Methodology:** Andrea J. Lund, Susanne H. Sokolow, Isabel J. Jones, Chelsea L. Wood, David López-Carr.

**Project administration:** Nicolas Jouanard, Anne-Marie Schacht, Simon Senghor, Gilles Riveau.

**Resources:** Nicolas Jouanard, Anne-Marie Schacht, Simon Senghor, Gilles Riveau.

**Supervision:** Susanne H. Sokolow, David López-Carr.

**Visualization:** Andrea J. Lund.

**Writing – original draft:** Andrea J. Lund.

**Writing – review & editing:** Andrea J. Lund, Susanne H. Sokolow, Isabel J. Jones, Chelsea L. Wood, Sofia Ali, Nicolas Jouanard, Gilles Riveau, Giulio A. De Leo, David López-Carr.

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
