## [Decision Letter · Decision Letter 0]

9 Apr 2021

Dear Ms. Lund,

Thank you very much for submitting your manuscript "Exposure, hazard and vulnerability all contribute to Schistosoma haematobium re-infection in northern Senegal" for consideration at PLOS Neglected Tropical Diseases. As with all papers reviewed by the journal, your manuscript was reviewed by members of the editorial board and by several independent reviewers. In light of the reviews (below this email), we would like to invite the resubmission of a significantly-revised version that takes into account the reviewers' comments. To me this reads as a mixed methods paper and could be sold more so on this aspect and the benefits that this provides, also helping to address the issues with smaller sample sizes, that would not be satisfactory for a purely quantitative study.

We cannot make any decision about publication until we have seen the revised manuscript and your response to the reviewers' comments. Your revised manuscript is also likely to be sent to reviewers for further evaluation.

Sincerely,

Poppy H L Lamberton

Deputy Editor

Anna Last

Associate Editor

Thanks for this interesting paper which we would be happy to see resubmitted after undertaking the recommended changes. To me this reads as a mixed methods paper and could be sold more so on this aspect and the benefits that this provides, also helping to address the issues with smaller sample sizes, that would not be satisfactory for a purely quantitative study.

Reviewer's Responses to Questions

**Key Review Criteria Required for Acceptance?**

**Methods**

-Are the objectives of the study clearly articulated with a clear testable hypothesis stated?

-Is the study design appropriate to address the stated objectives?

-Is the population clearly described and appropriate for the hypothesis being tested?

-Is the sample size sufficient to ensure adequate power to address the hypothesis being tested?

-Were correct statistical analysis used to support conclusions?

-Are there concerns about ethical or regulatory requirements being met?

Reviewer #1: yes

Reviewer #2: (No Response)

Reviewer #3: Overall, the methods are appropriately chosen in response to the study questions/objectives. The only minor issues is related to interview desribed in the study. Only two interviews/village with a total of 30 interview were conducted – what are potential implications with the analysis with such a few samples? Also unclear how 210 activity-specific observations were conducted?

Reviewer #4: Methods are explained in detail. Some limitations with the use of existing data collected from the communities is discussed.

**Results**

-Does the analysis presented match the analysis plan?

-Are the results clearly and completely presented?

-Are the figures (Tables, Images) of sufficient quality for clarity?

Reviewer #1: The authors ignore many relevant papers that have been published, which are given in a sticky note on the returned ms. They completely ignore the fact that very few snails have to be infected to result in a high (>87%) of school age children. They record the number of times that the children were in the water but they do not record the time of day they had water contact, which is extremely important.

Reviewer #2: (No Response)

Reviewer #3: Well done!

Reviewer #4: The results are clearly outlined and address the aims (integrated consideration of hazards, exposure and vulnerability).

SOme of the tables and figures could/should have acronyms described more clearly in headers and/or legends.

**Conclusions**

-Are the conclusions supported by the data presented?

-Are the limitations of analysis clearly described?

-Do the authors discuss how these data can be helpful to advance our understanding of the topic under study?

-Is public health relevance addressed?

Reviewer #1: There conclusions ignore important literature and thus they make sweeping statements that are not supported by other studies. Over all they have an excellent paper, they just need to add additional references.

Reviewer #2: (No Response)

Reviewer #3: Overall well done.

Reviewer #4: Conclusions are sounds. The discussion of weaknesses in the data at every step is commendable but detracts from the significance of their findings.

The authors clearly discuss what could be taken away from this study for efforts to disrupt the parasite lifecycle

**Editorial and Data Presentation Modifications?**

Reviewer #1: (No Response)

Reviewer #2: (No Response)

Reviewer #3: Accept

Reviewer #4: See comments below

**Summary and General Comments**

Reviewer #1: The title indicates it is a study of re-infection, which is misleading. In order to measure re--infection they would have to identify infected children, treat them, re-test them, and then have them exposed to water and determine re-infection

Reviewer #2: In this paper, Lund and colleagues address the interesting question of the interaction between hazard, exposure and vulnerability associated with infection and infection intensity of S. haematobium in Senegal. They found that infection presence was mostly driven hazard and vulnerability, while infection intensity is determined by exposure and hazard. I think this is an interesting paper that deserves to be published in PLoS NTDs when my comments would have been addressed.

I see two main concerns with the statistical analysis. First, they select one variable for each component (exposure, vulnerability and hazard), and then perform model selection to identify which one is the most parcimonious. Doing things like that can introduce biases. Indeed, if two variables included in their "hazard" category could explain the whole pathogen presence (or intensity), this cannot be identified because only one variable by component is selected and confronted to the others. I understand the rationale behind that, but I think it would be more robust to perform the analysis without such variable grouping.

Second, in the case where they select only one variable by group, I think including the interactions between variables could be interesting, since exposure and hazard could definitely interact between them.

On a minor note, from table 5, I don't understand the difference between model 16 and 19.

Reviewer #3: (No Response)

Reviewer #4: This research article highlights the importance of environmental hazards, exposure and vulnerability on the risks of urogenital schistosomiasis affecting school-aged children in a region of northern Senegal. Their findings extend work that has primarily focused on the demographic features of a population that affect the risk of infection. Consideration for each component of risk seems highly logical but rarely reported in an integrated way as they have reported here. This is a key, novel feature of this article (in my opinion). Discussion highlights several weaknesses in the approach used herein. Although this has limited the power of the analyses (as mentioned in the discussion), it also highlights that there is a lot more work to do in planning future epidemiological work. Overall, the methods look quite comprehensive. I was unable to review the DRYAD information using the url provided for dryad.s7h44j15p. Is it publicly available? You mention three studies that were undertaken to obtain the data used herein. Some abbreviation of previously published methods may be warranted here. Please make sure they are cited, similar to the PNAS manuscript (reference 18). This will make it clear what methods are unique to this manuscript. 

I have some other suggestions that I think are worth considering.

The introduction suggests a broad range of hazards should be considered, including floods and extreme weather events. Despite this being a focus in the introduction, it wasn’t a focus on the analysis and no data reported on extreme/catastrophic events in the region studied during the time of sampling. For this reason, I suggest simplifying the introduction to focus on the risks identified in a more usual seasonal cycle in the region (as presented in the results and discussed).

Narrative in the introduction can be confusing. I would consider starting with schistosomiasis or significantly reduce the background at the start of the introduction. I don't think an extended justification for looking at risk (exposure, hazard, and vulnerability) is required (start of the introduction). Most readers would be aware of this topic. Brief explanation would be better with an introduction that is focussed on the subject (schistosomiasis).

Lines 144 to 147. Please make it clear in the introduction why demographic features were included as control here. It is in the methods, results and discussion, but the approach could be better presented in the last paragraph of the discussion.

Line 222 to 223. Something missing in this sentence

Lines 529 to 532.Please include references to support these statements.

Line 618 and Line 648. Manuscript is well written but please check for grammatical errors throughout 

Discussion. Would it be better to incorporate the limitations of this study into one section and discuss how this can be resolved in the future. It feels repetitive at times and detracts from the significant results obtained in this study.

PLOS authors have the option to publish the peer review history of their article (what does this mean?). If published, this will include your full peer review and any attached files.

Reviewer #1: Yes: Jay Richard Stauffer, Jr.

Reviewer #2: No

Reviewer #3: No

Reviewer #4: No
---

## [Decision Letter · Decision Letter 1]

25 Aug 2021

Dear Ms. Lund,

Thank you very much for submitting your manuscript "Exposure, hazard, and vulnerability all contribute to Schistosoma haematobium re-infection in northern Senegal" for consideration at PLOS Neglected Tropical Diseases. As with all papers reviewed by the journal, your manuscript was reviewed by members of the editorial board and by several independent reviewers. The reviewers appreciated the attention to an important topic. Based on the reviews, we are likely to accept this manuscript for publication, providing that you modify the manuscript according to the review recommendations. 

There are very few changes that need to be made to the manuscript. The reviewer 1 has attached a ms with some edits. In addition please include some additional minor changes from me and then I am sure the manuscript will be accepted straight away. It is a lovely paper and I really enjoyed reading it again.

Sincerely,

Poppy H L Lamberton

Deputy Editor

Anna Last

Associate Editor

Reviewer's Responses to Questions

**Key Review Criteria Required for Acceptance?**

**Methods**

-Are the objectives of the study clearly articulated with a clear testable hypothesis stated?

-Is the study design appropriate to address the stated objectives?

-Is the population clearly described and appropriate for the hypothesis being tested?

-Is the sample size sufficient to ensure adequate power to address the hypothesis being tested?

-Were correct statistical analysis used to support conclusions?

-Are there concerns about ethical or regulatory requirements being met?

Reviewer #1: yes

Reviewer #2: (No Response)

Reviewer #3: As suggested in reviews for the 1st submission, the methods are overall conceived and designed. Minor issues were appropriately addressed in the revision.

**Results**

-Does the analysis presented match the analysis plan?

-Are the results clearly and completely presented?

-Are the figures (Tables, Images) of sufficient quality for clarity?

Reviewer #1: yes

Reviewer #2: (No Response)

Reviewer #3: Well done!

**Conclusions**

-Are the conclusions supported by the data presented?

-Are the limitations of analysis clearly described?

-Do the authors discuss how these data can be helpful to advance our understanding of the topic under study?

-Is public health relevance addressed?

Reviewer #1: yes

Reviewer #2: (No Response)

Reviewer #3: Yes.

**Editorial and Data Presentation Modifications?**

Reviewer #1: There are many places in the ms that nouns are used as adjectives. I marked many of these, but the authors should do a global search. One cannot use scientific names as adjectives! The introduction should be expanded slightly to indicate that there are many types of schistosomes that infect other mammals and birds. Should clarify that the paper deals with urinary schistosomes of humans, but there are also schistosomes that infect the bowels of humans.

Reviewer #2: (No Response)

Reviewer #3: (No Response)

**Summary and General Comments**

Reviewer #1: This is an excellent study and should be published.

Reviewer #2: I am happy with author's replies and therefore recommend acceptance

Reviewer #3: The authors have done a good job in addressing the reviewer's comments.

PLOS authors have the option to publish the peer review history of their article (what does this mean?). If published, this will include your full peer review and any attached files.

Reviewer #1: Yes: Jay R. Stauffer, Jr.

Reviewer #2: Yes: Benjamin Roche

Reviewer #3: No

Editor Poppy Lamberton comments:

In line 61 in the authors summary please change from ‘is acquired, where increasing numbers of worms lead to increasing severity of disease.’ To ‘is acquired. Increasing numbers of worms is known to be positively associated with increasing severity of disease.’ As you don’t actually measure the severity of disease in your study.

Intro line 71: respond to, or recover from,…. Please add these two commas.

Line 126 change to hotspots and line 727, as for most of the references for it.

Methods: line 190, please clarify that you mean urine on two days at each time point, or a single urine sample at two time points? I assume the former, but good to clarify.

Line 235: Please change schistosome infections to Schistosoma haematobium as that was what you tested for.

Lines 237: please clarify at what level you mean the medians, a cross the whole cohorts at each time point? Or across multiple samples from each person at each timepoint? And you can justify the median also as I assume the data were not normally distributed? Likely negative binomial distribution?

Results

Line 432: best fit: I think this is an example of where you use a noun rather than the verb. Maybe rephrase to was the best fit for the s.h presence data. 

Discussion:

Line 607: found ‘the’ area of snail habitat to be a…: please add either a ‘the’ or inverted commas: found ‘area of snail habitat’ to be a …

Line 616: change fit to fitted.

Line 617: found proximity to water access points: please write either found the proximity to water access points…. Or found ‘proximity to water access points’ as above

Line 623: fit to fitted

Line 662: represented access to, and use of, water and sanitation 

Line 666: in ‘the’ highest and lowest SES quintiles

Line 716: household level, most contribute…. Please change to household level, contribute most…. And for line 718

Line 718: change parasites to S. haematobium parasites

Figure Files:

Data Requirements:

Reproducibility:

References

---

## [Editor Report · Decision Letter 2]

10 Sep 2021

Dear Dr. Lund,

We are pleased to inform you that your manuscript 'Exposure, hazard, and vulnerability all contribute to Schistosoma haematobium re-infection in northern Senegal' has been provisionally accepted for publication in PLOS Neglected Tropical Diseases.

Best regards,

Poppy H L Lamberton

Deputy Editor

Anna Last

Associate Editor

---

## [Editor Report · Acceptance letter]

30 Sep 2021

Dear Dr. Lund,

We are delighted to inform you that your manuscript, "Exposure, hazard, and vulnerability all contribute to Schistosoma haematobium re-infection in northern Senegal," has been formally accepted for publication in PLOS Neglected Tropical Diseases.

Best regards,

Shaden Kamhawi

co-Editor-in-Chief

Paul Brindley

co-Editor-in-Chief
